# Regional Yield Estimation for Sugarcane Using MODIS and Weather Data: A Case Study in Florida and Louisiana, United States of America

**Shun Hu** [1], **Liangsheng Shi** [2,*], **Yuanyuan Zha** [2] and **Linglin Zeng** [3]

1    Hubei Key Laboratory of Yangtze River Basin Environmental Aquatic Science, School of Environmental Studies & State Key Laboratory of Biogeology and Environmental Geology, China University of Geosciences, Wuhan 430074, China
2    State Key Laboratory of Water Resources and Hydropower Engineering Sciences, Wuhan University, Wuhan 430072, China
3    College of Resources and Environment, Huazhong Agricultural University, Wuhan 430070, China
*    Correspondence: liangshs@whu.edu.cn

**Abstract:** Sugarcane is an important sugar source in America and is mainly planted in the states of Florida and Louisiana. The purpose of this study was to predict the sugarcane yield in these two states from 2008 to 2016. Three statistical sugarcane yield models (i.e., the CNDVI, K–M, and SiPAR models) were applied to predict yield in America, using remote sensing and weather data. To verify the robustness of models, model parameters obtained in places (i.e., Reunion Island and Southwestern China) far away from America were used. The results showed that the SiPAR model outperformed the CDNVI and K–M models for yield prediction. Solar radiation was an important constraint factor to ensure the statistical model's robustness under different conditions. The CNDVI model had the lowest robustness because of the absence of solar radiation, although it could reflect the yield trend to some extent. The K–M model failed to predict the low sugarcane yield, owing to the lack of consideration of temperature and soil water stresses. Florida had a low sugarcane yield in the west and southwest; however, Louisiana had high sugarcane yield in the same directions. This study demonstrated the robustness of the SiPAR model and investigated the sugarcane yield status in Florida and Louisiana. It can be a reference for similar studies in the future.

**Keywords:** sugarcane; regional yield estimation; model comparison; SiPAR model; weather factors





## 1. Introduction

Sugarcane is a tall perennial grass that thrives in tropical and subtropical climates. It is an important sugar source and accounts for roughly 75% of the world's sugar [1,2]. America is one of the top ten sugarcane-producing countries and the fifth largest consumer of sugar. Sugarcane provides about 45% of total domestic sugar in America. The heart of the America sugarcane industry spans Louisiana and Florida, which produced approximately 38 and 54% of the total sugarcane production in 2016, respectively [3]. Sugarcane contributes significantly to state economies and impacts local social and environmental issues. The accurate prediction of sugarcane yield is important for the sugarcane farmer, sugar manufacturer, market participant, and policymaker.

Process-based crop models and statistical models provide approaches to predicting sugarcane yield. Several sugarcane crop models, such as DSSAT-Canegro [4–6], APSIM-Sugarcane [7], and QCANE [8], need accurate model inputs (e.g., weather data, irrigation, and fertilizer) and parameters to precisely simulate growth and predict yield by imitating the interactions between the soil, atmosphere, and plant. However, uncertainties in the model structure, parameters, and inputs often limit the application of the crop model, particularly in regional or global scales [9–11]. Although a data assimilation method has

proved to be useful for improving yield prediction, the performance of data assimilation will deteriorate owing to data inadequacy (e.g., lack of root zone soil moisture observation) under suboptimal conditions, such as the existence of water stress [11–14]. Thus, for regional yield prediction, approaches based on the crop models still face challenges.

With the development of remote-sensing technology, statistical models based on the remotely sensed data have been used to predict crop yield at regional scales. They commonly relate the yield to vegetation indices (e.g., Normalized Differences Vegetation Index (NDVI) [15] and Enhanced Vegetation Index (EVI) [16]) calculated from remotely sensed data. Several linear and nonlinear statistical methods have been developed and applied to predict yield (e.g., the ones in References [17–22]). Statistical models require fewer datasets and are easy to operate. However, Abdel-Rahman and Ahmed and Basso et al. determined that most statistical models lack an explanation of their mechanism and are limited when providing information outside the range of values for which the model is parameterized [23,24]. It was suggested that introducing physiological concepts would be helpful to generalize statistical models and improve their reliability.

Considering that crop yield is closely related to the intercepted Photosynthetically Active Radiation (iPAR), Monteith proposed a semi-physical crop-yield prediction model using iPAR with the two parameters of harvest index (HI) and radiation use efficiency (RUE) [25]. Kumar and Monteith further modified this model (called the K–M model in this study) to make it applicable over a regional scale by using solar radiation and remotely sensed NDVI data [26]. Morel et al. used this modified model to predict sugarcane yield with the sugarcane parameters [27]. However, the performance was inferior to the empirical model developed with cumulated NDVI data (called the CNDVI model in this study). It was found that, under suboptimal growing conditions, the application of the models from Reference [25] is limited to the assumption of constant HI and RUE, which are affected by nitrogen, water, and temperature stresses [28,29]. Hu et al. proposed a new sugarcane yield prediction model called the SiPAR model [30]. The SiPAR model links iPAR to yield by multiplying an enhanced weighting factor of α calculated from the daily NDVI over the entire sugarcane growing season; α reflects the stem potential growth rate. The SiPAR model is well validated by using field experimental data and obtains a better performance than the traditional statistical models and data assimilation [30]. In this study, the SiPAR model was applied to predict the sugarcane yield in the states of Florida and Louisiana. The performance of the SiPAR model was verified by using statistical yield data at the state and county levels and compared with the K–M and CNDVI models.

The objectives of this study were to (1) predict the sugarcane yield by combining data from different sources, (2) compare different statistical models for mapping the sugarcane yield in Florida and Louisiana from 2008 to 2016, and (3) analyze the spatial pattern of the sugarcane planting area and yield in the two states. The study area and required data are described in Section 2. The three statistical models created for yield mapping and performance evaluation methods are introduced in Section 3. The results and discussions of predicted sugarcane yield are given in Section 4. Finally, Section 5 provides the main conclusions of this study.

## 2. Study Area and Data Collection

### 2.1. Study Area

Florida and Louisiana were the study areas (Figure 1). These two states are the main sugarcane producers in America. As shown in Figure 1, sugarcane is planted mainly in the southeastern parts of the states. In 2016, the sugarcane planting area was 168,754.1 ha in Florida, which was slightly smaller than Louisiana with 174,419.7 ha. However, the sugarcane planting distribution in Florida was more concentrated than in Louisiana. The average elevations of sugarcane planting area were 13.3 and 11.7 m in Florida and Louisiana, respectively. The climate in these two states is subtropical monsoon humid climate. The highest temperature is in the summer accompanied by the most precipitation. The annual

mean temperature and the total precipitation were 24.1 °C and 1176 mm in Florida and 20.9 °C and 1438 mm in Louisiana, respectively.

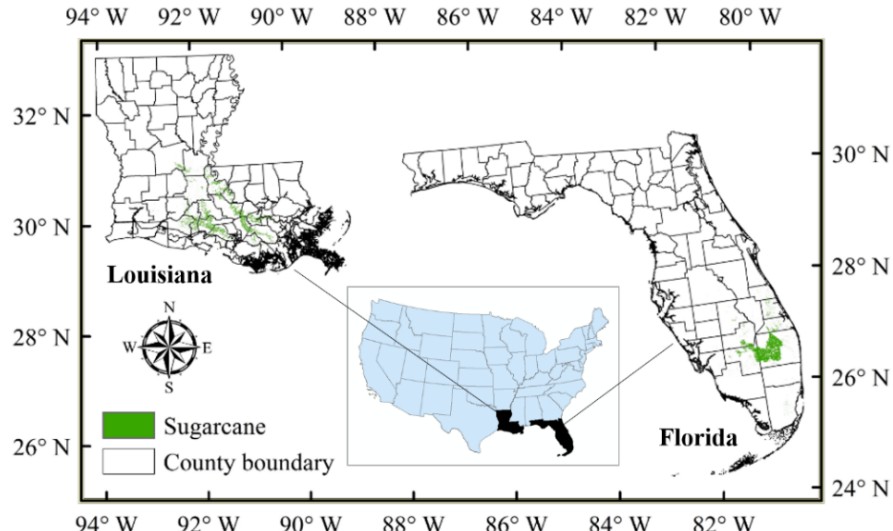

**Figure 1.** Study areas and the locations of sugarcane planting areas.

### 2.2. Data Collection

In this study, we predicted the sugarcane yield by using the SiPAR, K–M, and CNDVI models from 2008 to 2016 (total of 9 years yield). Daily NDVI, leaf area index (LAI), air temperature, and SR data are needed to complete models. In the following subsection, detailed descriptions of data used in this study are presented.

### 2.2.1. NDVI and LAI Data

The NDVI data were obtained from the MODIS/Terra satellite with a spatial resolution of 250 m and a temporal resolution of 16 days (retrieved 17 September 2020, from https://modis.gsfc.nasa.gov/) (the corresponding product is MOD13Q1). The NDVI was calculated by using the following equation: (NIR − Red)/(NIR + Red). NIR and Red are the surface reflectances of near-infrared and red bands, respectively. In the SiPAR model, daily NDVI data were needed. To obtain the daily NDVI, a shape-preserving piecewise cubic interpolation (phcip) method was used to interpolate the daily data from the remotely sensed data with the nominal date of MOD13Q1 product. Note that the highest spatial resolution of LAI data from the MODIS website was 500 m, which was different from the NDVI data. Thus, we did not directly use the LAI data from MODIS to complete the SiPAR model. A hyperbolic formula developed by Reference [30] was used to calculate the LAI from the NDVI at 250 m resolution. In this formula, the minimum and maximum NDVI values were 0.2 and 0.91, respectively. Two fitting parameters of $k$ and $d$ were obtained by using the field experimental data in Reference [30]. The same values of $k$ and $d$ were used in this study.

### 2.2.2. Sugarcane Statistical Planting Area, Yield, and Production Data

State- and county-level statistical data of sugarcane planting area, yield, and production were obtained from the National Agricultural Statistics Service (NASS) (retrieved 19 September 2020, from https://www.nass.usda.gov/). Supplementary Table S1 presents the state- and county-level statistical data. Note that state-level data included two parts: sugar plant and seed plant. The sugar plant of sugarcane is used for producing sugar in the mill. The seed plant of sugarcane is used for sowing in the following season. The seed planting area accounts for approximately 5% of the total planting area from Supplementary Table S1. However, county-level data include only the dataset of the sugar plant. Because the seed plant data were missing from the county-level data, the predicted production at

the county level may be higher than the statistical data. Because of the low proportion of seed plants, the predicted yield at the county level may not be significantly influenced.

### 2.2.3. Sugarcane Planting Area Map

Historical sugarcane planting area maps from 2008 to 2016 were obtained from the Cropland Data Layer (CDL) data of the National Agricultural Statistics Service (NASS) (retrieved 19 September 2020, from https://www.nass.usda.gov/). The original spatial resolution of the CDL was 30 m, corresponding to the Landsat satellite. To match the spatial resolution of NDVI data, the original sugarcane maps from the CDL were resampled with 250 m spatial resolution. However, there were two main sources of uncertainties for the CDL data. On the one hand, CDL data indicated that the land use for a special crop mainly based on remote sensing, which is more or less uncertain. On the other hand, the CDL maps included the road in the sugarcane planting area and other non-sugarcane fields, particularly in the resampled maps with 250 m spatial resolution. Figure 2 shows the sugarcane planting area of the two states from the CDL maps with the original 30 m and the resampled 250 m spatial resolutions against the statistical data (described in Section 2.2.2 and shown on the *x*-axis) from 2008 to 2016. It was apparent that both the original and resampled CDL maps overestimated the sugarcane planting area. Additionally, in the sparse sugarcane planting area, the resampled CDL maps may have large errors. Thus, it was necessary to first extract the sugarcane planting area from the CDL maps based on the temporal pattern of NDVI data. Then the sugarcane yield prediction performance was analyzed under different planting densities.

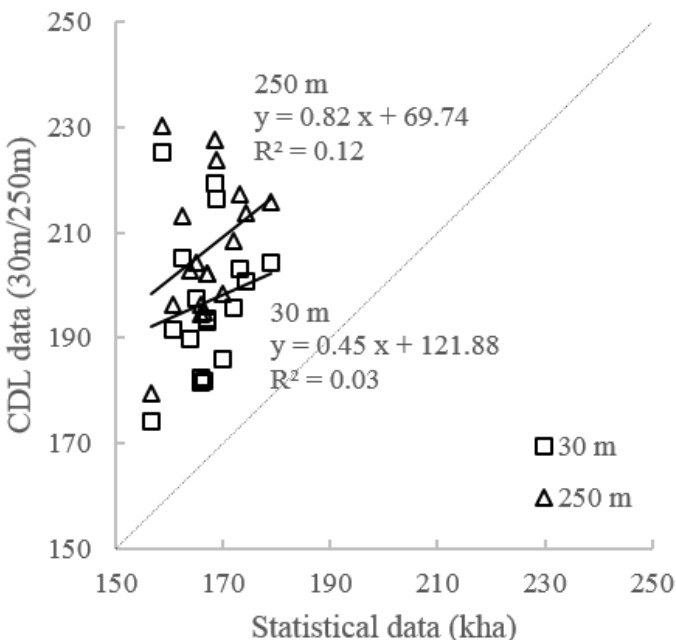

**Figure 2.** The sugarcane planting area from CDL maps with original 30 m and resampled 250 m spatial resolutions against statistical data from 2008 to 2016.

### 2.2.4. Meteorological and Soil Water Data

(1)　Solar radiation (SR) data

The regional SR data used in this study were Global Horizontal Irradiance (GHI) data downloaded from the NSRDB (retrieved 21 September 2020, from https://nsrdb.nrel.gov/). The NSRDB provided solar irradiance at a 4 km horizontal resolution for each 30 min interval from 2008 to 2016. The NSRDB data were computed by the National Renewable Energy Laboratory (NREL)'s Physical Solar Model (PSM) with satellite-based solar radiation. The global mean percentage bias for GHI was within 5%. To obtain SR

maps with the 250 m spatial resolution from the NSRDB GHI data, the Inverse Distance Weighted (IDW) interpolation method was used.

Considering that the SR maps from the NSRDB had a bias against the data from the ground weather stations, we hoped to correct it by using ground observation. The Florida Automated Weather Network (FAWN) includes 44 weather stations to monitor the weather data. The two stations of Belle (80.63°W, 26.66°N) and Clewiston (81.05°W, 26.74°N) are located in the sugarcane planting area. They provide long-term solar radiation observation. Comparing the SR from the NSRDB and from these two stations, the ratios of $SR_{NSRDB}/SR_{BC}$ were calculated from 2008 to 2016 (Figure 3). $SR_{NSRDB}$ is the annual sum of SR from the NSRDB at the Belle and Clewiston stations. $SR_{BC}$ is the annual sum of SR from the Belle and Clewiston stations. It can be seen that there was an obvious overestimation of SR by the NSRDB. As shown in Figure 1, the sugarcane planting area was concentrated in Florida; we believe it was reasonable to correct the SR maps from the NSRDB by using the ratios in Figure 3 for each year. However, in Louisiana, the sugarcane was sparse, and there were no appropriate weather stations to correct the data from the NSRDB.

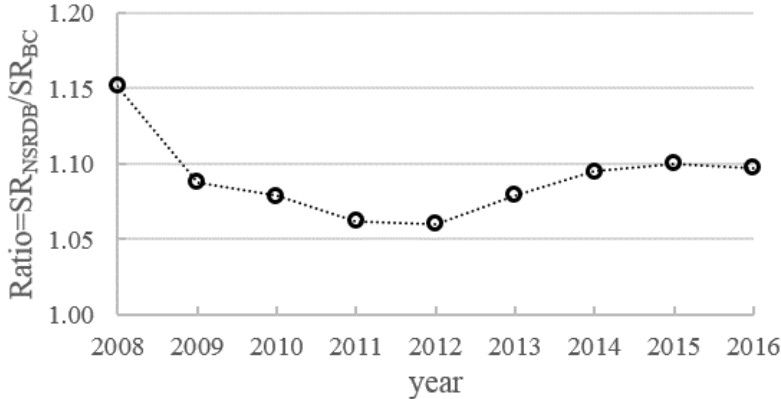

**Figure 3.** The ratios of $SR_{NSRDB}/SR_{BC}$ from 2008 to 2016.

(2)   Temperature data

The air-temperature data used in this study were parts of the forcing data for Phase 2 of the North American Land Data Assimilation System (NLDAS-2). The data can be downloaded from https://ldas.gsfc.nasa.gov/nldas (retrieved 23 September 2020) and have hourly temporal resolution. Their original spatial resolution was 0.125° (about 12.5 km at the equator). To match the spatial resolution of NDVI data, the temperature data were interpolated to 250 m, using the IDW method. Daily mean temperatures were calculated from the hourly temperature data.

## 3. Methodology

In this study, three statistical models (i.e., the CNDVI, K–M, and SiPAR models mentioned in Section 1) were used to predict the sugarcane yield in Florida and Louisiana. These models required knowing the length of the sugarcane growth period. To judge the start and end times of sugarcane growth, a method based on the temporal evolution of NDVI data was developed (Section 3.1). In addition, as mentioned in Section 2.2.3, it was necessary to develop a method to extract the most likely pixels of where the sugarcane was planted (Section 3.2). The procedures of CNDVI, K–M, and SiPAR models are given in detail in Section 3.3. Finally, the evaluation method for the performance of the models is presented in Section 3.4.

### 3.1. The Length of Sugarcane Growth Period

In Florida and Louisiana, the sugarcane harvest date is from October to May of the following year. In August, the sugarcane is planted and not harvested. This characteristic of sugarcane planting is helpful in judging the start and end times of sugarcane growth.

Figure 4 shows the temporal evolution of NDVI data of a representative pixel from 2007 to 2009. The time in P1 is from 1 October 2007 to 31 July 2008, and the time in P2 is from 1 August 2008 to 31 May 2009. According to the characteristics of sugarcane planting, the starting and ending times must be located in P1 and P2, respectively. In this study, we determined that the time corresponding to the minimum NDVI in P1 was the starting time, and the time corresponding to the minimum NDVI in P2 was the ending time for the growth season in 2008. Using this method for each pixel, the length of the growth period, as well as the start and end times, was obtained.

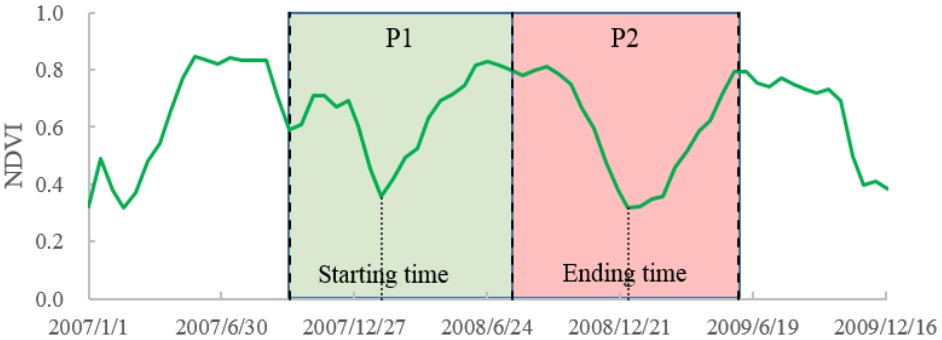

**Figure 4.** The temporal evolution of a representative pixel.

### 3.2. Sugarcane Planting Area Extraction

To eliminate the non-sugarcane pixels in the 250 m resolution CDL maps, the temporal evolution of NDVI data were used. Firstly, as shown in Figure 4, if the starting time was before 1 October or the ending time after 31 May in one pixel, the crop type in this pixel was not sugarcane. In addition, when the starting and ending times were obtained by using the method in Section 3.1, the length between the starting and ending times and average and standard deviation of the NDVI between the starting and ending times were calculated. If the length was less than 180 days, the average NDVI was larger than 0.75 or lower than 0.4, or the standard deviation was smaller than 0.13 of one pixel, the crop type of this pixel was not sugarcane. After completing these processes, the sugarcane planting area was extracted. Figure 5 shows the extracted sugarcane planting area against the 250 m spatial resolution statistical data of the two states. It can be seen that, in only a few years (2012 and 2013 for Louisiana) were the sugarcane planting areas obviously underestimated.

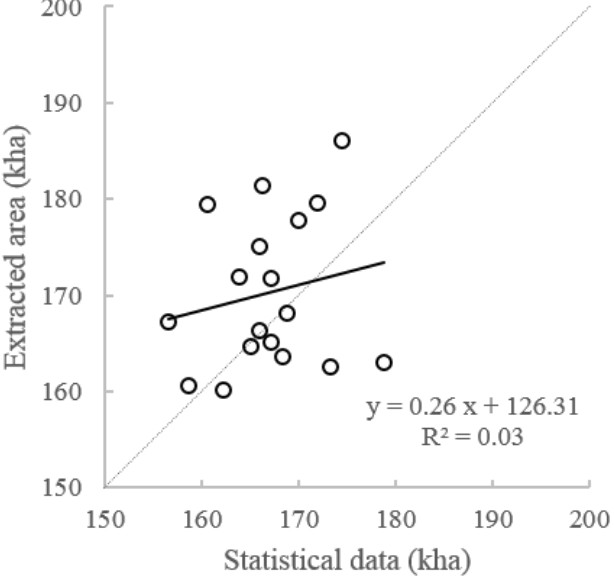

**Figure 5.** The extracted sugarcane planting area against the 250 m spatial resolution statistical data.

*3.3. Sugarcane Yield Prediction Models*

Three yield prediction models (i.e., the CNDVI, K–M, and SiPAR models) were used to map the sugarcane yields in Florida and Louisiana. Air mean temperature and NDVI data were needed to implement the CNDVI model. NDVI and SR data were necessary for the K–M and SiPAR models to predict yield. In the following subsections, the procedures of these three models are described in detail.

### 3.3.1. CNDVI Model

In Reference [27], Morel et al. developed a satellite-based sugarcane yield prediction model (i.e., CNDVI model) and applied it to Reunion Island. The core of the CNDVI model is linking the growing-season-integrated NDVI with the sugarcane yield:

$$\widehat{SY} = k{\cdot}GSiNDVI_L + b \tag{1}$$

where (SY) $\widehat{SY}$ is the predicted sugarcane yield; $GSiNDVI_L$ is the growing season-integrated NDVI of the entire growth period; and $k$ and $b$ are the model parameters, which were 0.033 and 14.09 in Morel et al. [27], respectively.

$GSiNDVI_L$ is calculated by cumulating the NDVI with the development of the thermal age. The thermal age was computed from the daily air mean temperature (AMT) with a base temperature (*Tbase*) of 12 °C for sugarcane in Reference [27].

$$GSiNDVI_l = \sum_{t=1}^{l} NDVI_t{\cdot}(AMT_t - Tbase) \tag{2}$$

where $AMT_t$ is the daily mean temperature at the $t$th day from the starting time, and $l$ is the total growing days from the starting time. When the time is at the ending time, $l$ becomes $L$ (the length of the entire growth period), and the $GSiNDVI_L$ is used in Equation (1) to predict sugarcane yield.

### 3.3.2. K–M Model

The original K–M model is a simplification of the Monteith model [25] and was developed by Kumar and Monteith [26]. It was based on the relationship between the *PAR* intercept efficiency and vegetation indices and can be applied with remotely sensed data. Researchers have used the K–M model to predict the aboveground biomass or yield of grain crop (e.g., References [31,32]). Morel et al. [27] was the first to use the K–M model to predict the sugarcane yield:

$$DM = RUE{\cdot}\varepsilon_b{\cdot}\sum_{t=1}^{L} fAPAR_t{\cdot}SR_t \tag{3}$$

$$fAPAR_t = 1.383{\cdot}NDVI_t - 0.333 \tag{4}$$

$$\widehat{SY} = p{\cdot}DM + q \tag{5}$$

where $DM$ is the dry matter production (g/m$^2$), and sugarcane yield is obtained by using Equation (5); $RUE$ represents the capacity of the plant to convert radiation into dry biomass, and it is set to 3.22 g/MJ; $fAPAR_t$ represents the capacity of a vegetation cover to intercept and adsorb incident radiation at the $t$th day, and it is calculated from remotely sensed $NDVI$ data, using Equation (4); $\varepsilon_b$ is the ratio of PAR to SR, and it is widely set to 0.5; and $p$ and $q$ are the model parameters, which were 0.018 and 3.64 in Morel et al. [27], respectively.

### 3.3.3. SiPAR Model

The SiPAR model was proposed in Reference [30], using an enhanced weighting factor ($\alpha$) from the NDVI to reflect the potential stem growth rate and using the intercepted PAR (iPAR) as a constraint factor:

$$\widehat{SY} = \beta \cdot \sum_{t=1}^{L} \alpha_t \cdot iPAR_t \tag{6}$$

$$\alpha_t = e^{\lambda \cdot \omega_t} - 1 \tag{7}$$

$$\omega_t = (NDVI_t - NDVI_{min})/(NDVI_{max} - NDVI_{min}) \tag{8}$$

$$iPAR_t = \varepsilon_b \cdot \left(1 - e^{-Ki \cdot LAI_t}\right) \cdot SR_t \tag{9}$$

where $\omega_t$ is the normalized NDVI at the $t$th day; $Ki$ is the light extinction coefficient and is set to 0.65; $NDVI_{min}$ and $NDVI_{max}$ are the minimum and maximum NDVI and set to 0.2 and 0.91, respectively; and $\lambda$ and $\beta$ are the model parameters, and their values are 1.85 and 16.39, respectively. Note that, in Equation (9), LAI data are needed to calculate the iPAR. In Reference [30], Hu et al. used a hyperbolic function to compute LAI from NDVI data for sugarcane:

$$LAI_t = \sqrt[5]{20 \cdot \omega_t/(1 - \omega_t)} \tag{10}$$

The SiPAR model has obtained high accuracy and strong robustness at field scale. It has the potential to be applied at the region scale by using remotely sensed data, and to the best of our knowledge, the SiPAR model has not yet been applied at the regional scale. This study applied the SiPAR model to predict the sugarcane yield in Florida and Louisiana and compare its performance with the CNDVI and K–M models.

Note that, in the above three sugarcane yield prediction models, the model parameter values in this study were taken from the research of References [27,30], without any changes. The parameter values in the CNDVI and K–M models were obtained by using the weather station and remotely sensed data in Reunion Island. The parameter values in the SiPAR model were obtained by using the field experimental data in Guangxi, Southwestern China. Although Florida and Louisiana are far away from Reunion Island and China and have different crop genotypes and environmental conditions, it was helpful to validate the models' universality by applying the existing parameters of these models.

### 3.4. Performance Evaluation Method

After the sugarcane yield was predicted by the above three yield models, the performances of these models were evaluated at both the state and county levels. The proportions (i.e., Rbias and Rrmse) of bias and root mean square error (RMSE) to averaged statistical data were utilized to judge the accuracy of the predicted yield. The slope and determination coefficient ($R^2$) between the predicted and statistical yields were used to evaluate the reproduction of the spatial pattern at the regional scale.

$$Rbias = \sum_{i=1}^{N} \left(\widehat{SY_i} - SY_i\right) \Big/ \sum_{i=1}^{N} SY_i \tag{11}$$

$$Rrmse = \sqrt{\sum_{i=1}^{N} \left(\widehat{SY_i} - SY_i\right)^2/(N-1)} \Big/ ASY \tag{12}$$

$$ASY = \sum_{i=1}^{N} SY_i/N \tag{13}$$

where $\widehat{SY_i}$ and $SY_i$ are predicted and statistical yields (t/ha), $N$ is the total number of statistical data, and $ASY$ is the averaged statistical data (t/ha).

### 4. Result and Discussion

Three sugarcane yield models (i.e., the CNDVI, K–M, and SiPAR models) were applied to predict the yields in Florida and Louisiana. In addition to the analysis of the model performance at the state and county levels, the model performance under different planting densities was also analyzed because the planting density will influence the extraction of

the sugarcane planting area using remotely sensed data. Palm Beach County has a higher planting density than other counties. The model performances were only compared in Palm Beach County because the consideration of high planting density resulted in relatively fewer errors in the extraction of the planting area using remotely sensed data, especially with coarse spatial resolution. In addition, the spatial patterns of sugarcane yield in Florida and Louisiana were analyzed based on the results of the SiPAR model.

### 4.1. Extracted Sugarcane Planting Area at County Level

Figure 6 shows the extracted sugarcane planting area at the county level. It can be seen that the spatial pattern was well reproduced. Combining Figure 6 and Supplementary Table S1, we see that there was a big variance in planting area among counties. Florida's Palm Beach County had the largest planting area and the highest planting density. However, in Louisiana, many counties had relatively smaller planting areas with sparse distribution. The sparse distribution of sugarcane planting areas commonly results in low extraction accuracy when using remote sensing. Thus, it is necessary and meaningful to analyze the yield prediction performance with different planting areas.

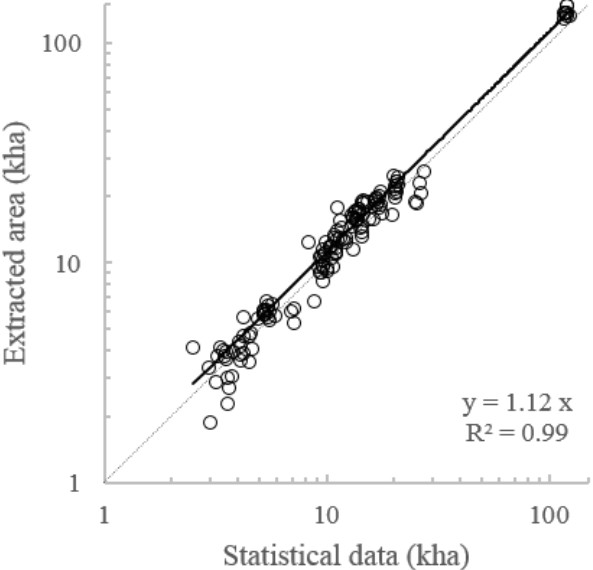

**Figure 6.** The county-level extracted sugarcane planting area against statistical data.

### 4.2. Sugarcane Yield Prediction

4.2.1. Prediction Performance at State Level

The CNDVI, K–M, and SiPAR models were applied to predict sugarcane yield in Florida and Louisiana. Figure 7 presents the prediction performance of the three models at the state level from 2008 to 2016. The three models were able to reflect the annual evolution of sugarcane yield on the consideration of slope and $R^2$ close to 1.

Compared with the K–M and SiPAR models, the CNDVI model obviously overestimated the yield for all years (Rbias = 17.26% against 7.66 and −3.30%). It also demonstrated the worst accuracy with the highest Rrmse (19.20% against 8.80 and 5.88%). In contrast, the SiPAR model obtained the best yield prediction of the three models. The inferior prediction of the CNDVI model indicated that the CNDVI model lacked universality under varied environments, despite it obtaining the best performance in Reference [27]. The reason may be that the parameter of Tbase in Equation (2) had a significant and directed influence on the calculation of GSiNDVI; however, this may change according to the location, cultivars, and phenological phases [33–35]. Many studies have used different values of Tbase in different positions with various cultivars to achieve their purposes (e.g., References [12,36,37]). The

higher accuracy of the K–M and SiPAR models may result from the use of solar radiation as the constraint factor to obtain relatively robust prediction.

**Figure 7.** The predicted yield against statistical yield at the state level.

### 4.2.2. Prediction Performance at County Level

To validate the reproduction of the spatial pattern of sugarcane yield, the predicted yield was analyzed by using the county-level data. Figure 8 shows the predicted sugarcane yield against statistical data at county level from 2008 to 2016 (a total 142 points). Similar to the results of predicted yield at the state level, the CDNVI model significantly overestimated the yield at the county level. The K–M and SiPAR models obtained better performances than the CNDVI model. Additionally, compared to the results in Figure 7, the performances of the K–M and SiPAR models deteriorated at the county level, especially in the counties of Louisiana. The reason may be that the sparse distribution of the planting area in Louisiana resulted in inaccurate extraction by using remote sensing data with a 250 m spatial resolution. To illustrate the influence of planting density on the model performance, Table 1 shows the yields for varied planting areas. Note that, in Florida and Louisiana, a large planting area commonly has high planting density over the region.

**Table 1.** The prediction performance under varied planting areas of county. F represents Florida, and L represents Louisiana.

| Model | Area kha | Number - | Slope - | $R^2$ - | Rbias % | Rrmse % |
|---|---|---|---|---|---|---|
| CNDVI | >2.5 | 27F, 115L | 0.84 | 0.50 | 13.23 | 17.19 |
| | >6.0 | 27F, 75L | 0.91 | 0.59 | 15.41 | 18.76 |
| | >10.0 | 21F, 64L | 0.97 | 0.64 | 14.95 | 17.80 |
| | >14.0 | 19F, 32L | 1.06 | 0.68 | 16.18 | 19.00 |
| | >18.0 | 16F, 9L | 1.10 | 0.81 | 20.54 | 22.33 |
| K–M | >2.5 | 27F, 115L | 0.58 | 0.40 | 6.38 | 12.40 |
| | >6.0 | 27F, 75L | 0.60 | 0.50 | 8.22 | 13.08 |
| | >10.0 | 21F, 64L | 0.63 | 0.53 | 7.97 | 12.16 |
| | >14.0 | 19F, 32L | 0.70 | 0.63 | 7.18 | 10.93 |
| | >18.0 | 16F, 9L | 0.66 | 0.74 | 9.25 | 11.97 |
| SiPAR | >2.5 | 27F, 115L | 0.83 | 0.58 | −7.72 | 12.21 |
| | >6.0 | 27F, 75L | 0.87 | 0.67 | −5.46 | 10.27 |
| | >10.0 | 21F, 64L | 0.89 | 0.66 | −5.62 | 10.11 |
| | >14.0 | 19F, 32L | 0.95 | 0.73 | −4.58 | 8.98 |
| | >18.0 | 16F, 9L | 0.92 | 0.80 | −0.78 | 6.58 |

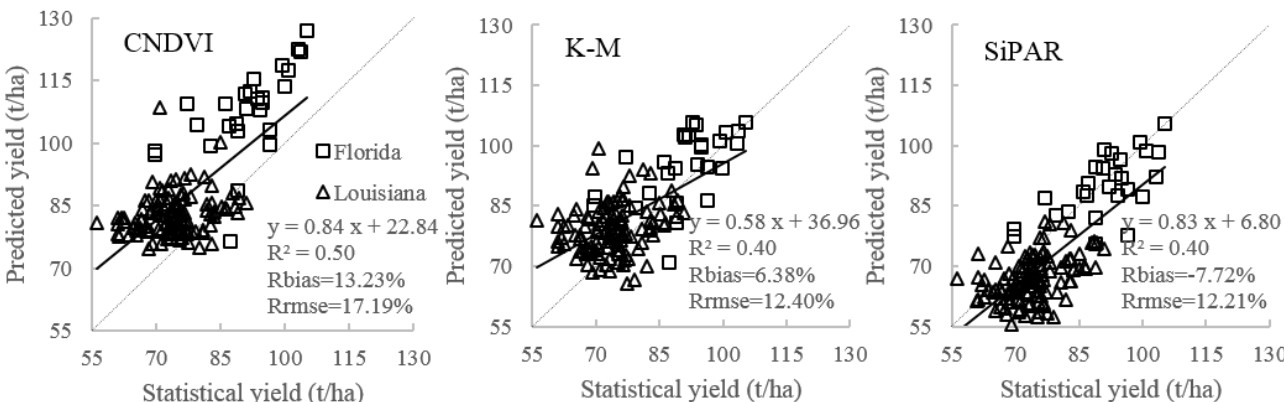

**Figure 8.** The predicted sugarcane yield against statistical data at the county level.

In Table 1, it is interesting to note that, with the increase in planting area in a county, the CNDVI model tended to more obviously overestimate the sugarcane yield. This demonstrated that the CNDVI was not able to predict sugarcane yield in Florida and Louisiana with the model parameters given by Reference [27], indicating that the CDNVI model was not appropriate for places without adequate ground data to calibrate the parameters. In addition, the performance of the K–M model seemed unrelated to the planting density. This reflected the inherent drawback of the K–M model, which always overestimated the yield. Fortunately, with the increase in planting density at the county level, the SiPAR model obtained higher accuracy in terms of slope, $R^2$, Rbias, and Rrmse. This demonstrated that, if the planting area is accurately extracted, the SiPAR is able to predict the yield accurately.

### 4.2.3. Prediction Performance in Palm Beach County

As indicated in the above discussion, the sparse distribution of sugarcane planting area will influence the evaluation of yield models. Beyond the analysis of model performance under different planting areas in Section 4.2.2, in this subsection, the models' performances is further discussed in regard to Palm Beach County (Figure 9). Palm Beach County accounts for approximately 75% of the sugarcane planting area and production in Florida. In Figure 9, we see that the sugarcane planting area is concentrated. The high planting density in Palm Beach resulted in the high accuracy of the extracted planting area from using remotely sensed data with 250 m spatial resolution. As mentioned before, the solar radiation data were corrected for Florida. Thus, comparing the model performance in Palm Beach County was the most reliable method.

The results of predicted yield from the CNDVI, K–M, and SiPAR models are also presented in Figure 9. The CNDVI model obviously overestimated the yield (Rbias = 22.35%; Rrmse = 23.96%), although it reproduced the annual pattern of yield from 2008 to 2016 well (slope = 0.95; $R^2$ = 0.86). Better results were obtained by the K–M and SiPAR models. However, the K–M model still overestimated the sugarcane yield, particularly when it was low. The reason for this phenomenon may be that the K–M model did not consider the stresses from soil water and temperature, and the influence of these stresses on model parameters was also ignored. Low sugarcane yield usually indicates that the sugarcane suffered stresses such as drought, waterlogging, low temperature, and heat wave. Fortunately, the SiPAR model reproduced the annual pattern of yield very well (slope = 0.95; $R^2$ = 0.96) and achieved the highest accuracy of the three models (Rbias = 2.43%; Rrmse = 3.18%). This demonstrated that the SiPAR model has strong robustness and is a reliable method of predicting sugarcane yield.

In the above analysis, the three tested models were not calibrated. For a more rigorous comparison, the cross-validation procedure was further implemented. Two-thirds of the data (6 years) were used as the calibration dataset and one-third of data (3 years) as the validation dataset in Palm Beach County. The constants of $k$ and $b$ in Equation (1), $p$

and $q$ in Equation (5), and $\beta$ in Equation (6) were recalibrated by using the calibration dataset. A total of 84 combinations of calibration and validation were completed. The validation results of the CNDVI, K–M, and SiPAR models in the cross-validation procedure are presented in Table 2. Comparing with Figure 9, we see that most of the evaluation metrics were improved through calibration, especially for the CNDVI and K–M models. However, the SiPAR model is still the optimal model. It is also found that the differences of evaluation metrics for the SiPAR model between that in Table 2 and in Figure 9 are small, thus further manifesting the robustness and reliability of the SiPAR model.

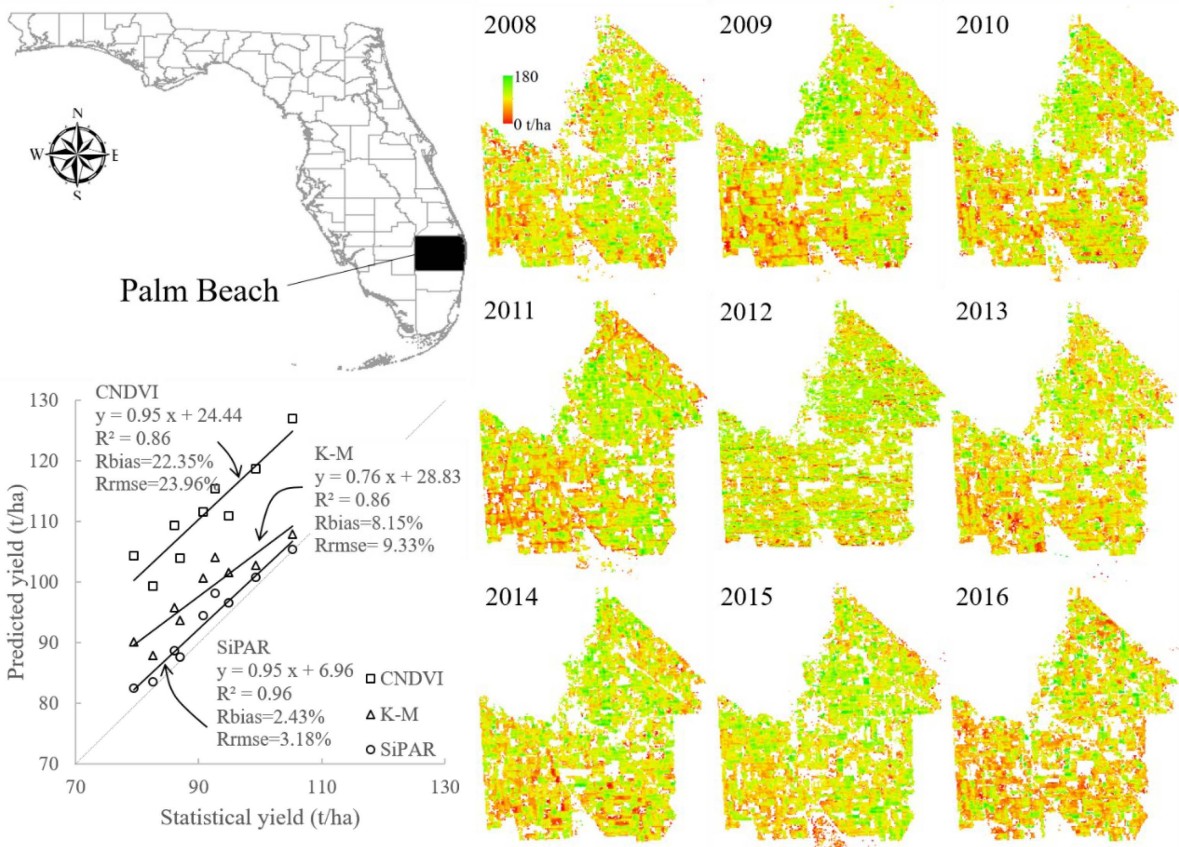

**Figure 9.** The spatial distribution of sugarcane yields in different years and the prediction performances of the CNDVI, K–M, and SiPAR models in Palm Beach County.

**Table 2.** The validation results of the CNDVI, K–M, and SiPAR models during cross-validation using the data in Palm Beach County.

| Model | Slope<br>- | $R^2$<br>- | Rbias<br>% | Rrmse<br>% |
|---|---|---|---|---|
| CNDVI | $0.86 \pm 0.39$ | $0.86 \pm 0.24$ | $-0.14 \pm 2.76$ | $2.93 \pm 1.78$ |
| K–M | $0.96 \pm 0.52$ | $0.87 \pm 0.19$ | $-0.38 \pm 2.90$ | $2.87 \pm 1.94$ |
| SiPAR | $0.97 \pm 0.22$ | $0.95 \pm 0.09$ | $0.04 \pm 1.35$ | $1.54 \pm 1.04$ |

*4.3. The Spatial Pattern of Sugarcane Yield*

From the state- and county-level results of predicted yield, we concluded that the SiPAR model was suitable and reliable for predicting yield on the regional scale. Figure 10 presents the sugarcane yield maps with 250 m resolution, using the SiPAR model, in Florida and Louisiana. Figure 10 directly reflects that the planting area was more concentrated in Florida than in Louisiana. Florida had a higher sugarcane yield than Louisiana. The low sugarcane yield was always located in the west and southwest of Florida; however, in Louisiana, the high sugarcane yield was always located in the same direction. From 2008

to 2016, the variation in sugarcane yield was lower in Louisiana than in Florida. When combining the results in Figure 7 with Figure 10, we noted that there may be inherent impact factors influencing the yield in Louisiana.

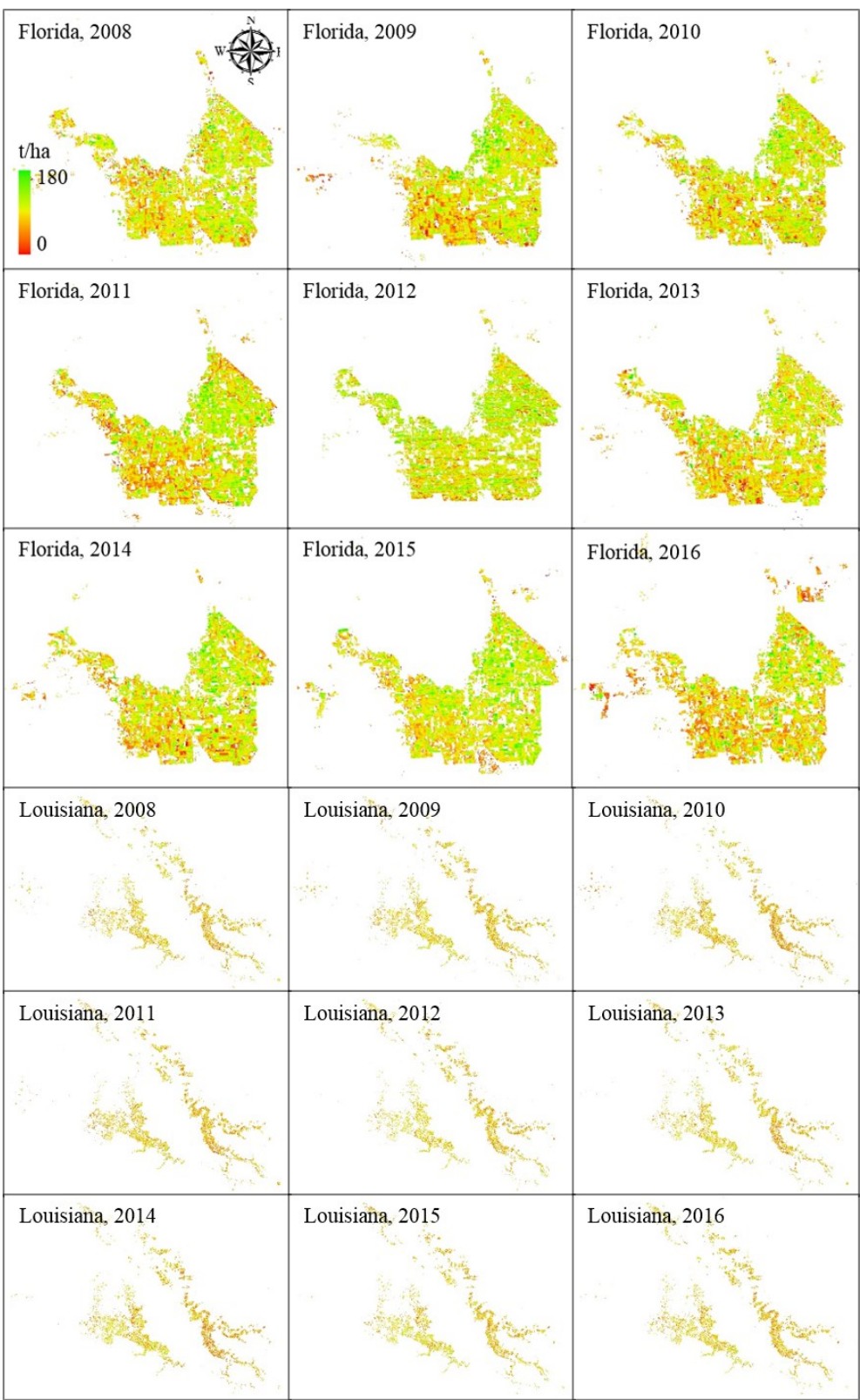

**Figure 10.** The predicted sugarcane yield maps using the SiPAR model in Florida and Louisiana from 2008 to 2016.

## 5. Conclusions

The purpose of this study was to predict the sugarcane yield and analyze the weather impact factors on sugarcane yields in Florida and Louisiana. Three existing models (i.e., the CNDVI, K–M, and SiPAR models) were applied to predict sugarcane yield. Note that the parameters of these models obtained outside our study area were not modified. This aimed to effectively verify the universality of the applied models.

Whether at the state or county level, the SiPAR model always outperformed the CNDVI and K–M models with/without the model recalibration. Although the CNDVI model reflected the annual and spatial patterns of sugarcane yield without the constraint of solar radiation, it could not obtain high accuracy and significantly overestimated the yield in our study area. On the other hand, the K–M and SiPAR models obtained better accuracy than the CNDVI model, owing to the constraint of solar radiation. However, without considering the stresses from soil water and temperature, the K–M model had inherent errors and consistently overestimated in the low yield zone. The SiPAR model not only well reproduced the annual and spatial patterns of sugarcane yield but also obtained the highest accuracy. This demonstrated that the SiPAR model is robust and can be applied to predict sugarcane yield outside the area where the model parameters were obtained. Note that, for the regional yield prediction, accurate solar radiation data and extracted planting area are necessary for the successful application of the SiPAR model.

This study demonstrated that the SiPAR model can be applied to predict the sugarcane yield without any modification of the model parameters. The methodology used in this study can be a reference for the research on sugarcane yield in other regions.

**Supplementary Materials:** The following supporting information can be downloaded at https://www.mdpi.com/article/10.3390/rs14163870/s1. Table S1: State- and county-level statistical data from NASS. For state-level data, the F and L following the county name represent Florida and Louisiana states, respectively.

**Author Contributions:** Conceptualization, S.H. and L.S.; methodology, S.H.; software, S.H.; validation, L.S., Y.Z. and L.Z.; formal analysis, S.H. and L.S.; investigation, S.H.; resources, S.H. and L.S.; data curation, S.H.; writing—original draft preparation, S.H.; writing—review and editing, S.H., L.S. and L.Z.; supervision, L.S.; project administration, S.H. and L.S.; funding acquisition, S.H. and L.S. All authors have read and agreed to the published version of the manuscript.

**Funding:** This research was funded by the Key Research and Development Program in Guangxi Grant (No. AB19245039), the Key Research and Development Program in Guangxi Grant (No. 2019AB20009), the Fundamental Research Funds for the Central Universities, China University of Geosciences (Wuhan) (No. 162301202679), and the Open Research Fund of Guangxi Key Laboratory of Water Engineering Materials and Structures Grant (No. GXHRI-WEMS-2020-06).

**Institutional Review Board Statement:** Not applicable.

**Informed Consent Statement:** Not applicable.

**Data Availability Statement:** Not applicable.

**Acknowledgments:** Thanks for the data provided by the National Agricultural Statistics Service.

**Conflicts of Interest:** The authors declare no conflict of interest.

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
