# Peer review of "Regional Yield Estimation for Sugarcane Using MODIS and Weather Data: A Case Study in Florida and Louisiana, United States of America"

_remotesensing, doi:10.3390/rs14163870_

Round 1

Reviewer 1 Report

The paper is well written and almost ready to be publish. The major question is related to the application of the statistical models for a different region without adjust their parameters. For example, it is clear that the b parameter of the CNDVI model is overestimated, resulting in an yield overprediction; i.e., the problem is not with the model, but with the parameters used (probably this model could have similar skill as the other models).  

More over, authors could do a first application without any calibration and argue that SiPAR presented the best result without local calibration, which is probably related to the inclusion of terms based on biophysical models. However, in a second step, the models should be recalibrated locally and their performance in relation to models without local calibration should be done.  

Author Response

Responses to Reviewer 1

The paper is well written and almost ready to be publish. The major question is related to the application of the statistical models for a different region without adjust their parameters. For example, it is clear that the b parameter of the CNDVI model is overestimated, resulting in a yield overprediction; i.e., the problem is not with the model, but with the parameters used (probably this model could have similar skill as the other models).

Moreover, authors could do a first application without any calibration and argue that SiPAR presented the best result without local calibration, which is probably related to the inclusion of terms based on biophysical models. However, in a second step, the models should be recalibrated locally and their performance in relation to models without local calibration should be done.

Response: Thank you very much for the comment. It is true that for statistical models applied in this study, if the model parameters are locally calibrated (e.g., adjusting the value of b parameter of the CNDVI model), these models will obtain similar performances (especially for the CNDVI model). This is the inherent characteristics of statistical models. However, for a robust statistical model, yield should be well predicted even though the model parameters are obtained using the data outside the region where the model will be applied. Local calibration cannot verify the spatial robustness of models because the model parameters have been changed. Therefore, we think it may be not necessary to conduct the local calibration in this study. We hope the reviewer could agree with us consideration.

Reviewer 2 Report

Manuscript for Remote Sensing (1812362) Regional yield estimation for sugarcane using remote sensing and weather data: a case study in Florida and Louisiana, America

Review Comments

I would like to thank the authors for the opportunity to review this research paper. The use of remote sensing, in particular public domain earth observing satellite imagery, to improve yield estimates and forecasting is a topic of great interest currently. The authors have presented the results of an improved model for sugar cane production in the USA, based on previous calibration in China.

My primary concern is with the execution of the final of the four stated objectives:

(1) predict the sugarcane yield by combining data from different sources, (2) compare different statistical models for mapping the sugarcane yield in Florida and Louisiana from 2008 to 2016, analyze the spatial pattern of sugarcane planting area and yield in the two states, and (4) discuss the weather impact factors for sugarcane yield.

The authors present the results to demonstrate that the SiPAR model predicted sugar cane yields more accurately than the other two models compared. However, the ‘analysis’ for the climate/weather impacts on sugar cane yield is not well executed or presented. I would suggest that the authors drop this objective and focus on the first three objectives of the study. It is a distraction, and it isn’t clear to me how the model results improve an analysis of climatic factors over the use of reported statistics.

In addition to the above comments, I have made comments throughout the manuscript (attached). I hope that the authors find these comments constructive.

Author Response

Responses to Reviewer 2

I would like to thank the authors for the opportunity to review this research paper. The use of remote sensing, in particular public domain earth observing satellite imagery, to improve yield estimates and forecasting is a topic of great interest currently. The authors have presented the results of an improved model for sugar cane production in the USA, based on previous calibration in China.

Thank you for the review and comments. These comments are all valuable and very helpful for revising and improving our paper as well as an important guide for our future research. We have considered your comments carefully and made revisions. The main revisions are all tracked in the revised manuscript while the responses to your comments are as follows.

My primary concern is with the execution of the final of the four stated objectives:

(1) predict the sugarcane yield by combining data from different sources, (2) compare different statistical models for mapping the sugarcane yield in Florida and Louisiana from 2008 to 2016, (3) analyze the spatial pattern of sugarcane planting area and yield in the two states, and (4) discuss the weather impact factors for sugarcane yield.

The authors present the results to demonstrate that the SiPAR model predicted sugar cane yields more accurately than the other two models compared. However, the ‘analysis’ for the climate/weather impacts on sugar cane yield is not well executed or presented. I would suggest that the authors drop this objective and focus on the first three objectives of the study. It is a distraction, and it isn’t clear to me how the model results improve an analysis of climatic factors over the use of reported statistics.

Response: Thank you for the comment very much. We have dropped the final objective and focused on the first three objectives.

In addition to the above comments, I have made comments throughout the manuscript (attached). I hope that the authors find these comments constructive.

Response: We have revised our manuscript according to them. They are very helpful to improve the manuscript.

Reviewer 3 Report

Summary

This manuscript compares three models predicting sugarcane yields. The models have been calibrated previously using data from other regions, and are evaluated here for prediction in Florida and Louisiana, thus testing their ability to extrapolate. Predictions from the ‘SiPAR’ model perform best here. The work seems rigorous, results well analysed and clearly presented, and I would suggest only a few minor corrections, listed below.

Details

Lines 193-205 : I think this is based on the bias of the sum or the mean over the two stations. But was the bias similar for both stations (in which case the bias correction is well justified), or was it much larger for one station than the other (in which case, the correction might be less justified by the data)?

Line 249 – Do you mean ‘standard deviation’ rather than ‘standard error’? Using ‘standard deviation’ would make more sense to me as a measure of the variation of NDVI within the given window (whereas standard error would be a measure of the precision of the estimated mean NDVI within the window). Similarly for other uses of ‘standard error’ (e.g. line 474 and other places).

Eq 12 : I found this equation a bit confusing – does it need brackets? (Probably should be around the ‘N-1’ , though I’m not sure why you use ‘N-1’ rather than ‘N’ as the denominator for RMSE).

Fig 9 : Planting area seems to be presenting in units of t/ha – I would have expected it to be in units of area, ie ha.

Author Response

Responses to Reviewer 3

This manuscript compares three models predicting sugarcane yields. The models have been calibrated previously using data from other regions, and are evaluated here for prediction in Florida and Louisiana, thus testing their ability to extrapolate. Predictions from the ‘SiPAR’ model perform best here. The work seems rigorous, results well analyzed and clearly presented, and I would suggest only a few minor corrections, listed below.

Response: Thank you for the review and comments. We have studied your comments carefully and made revisions. The main revisions are all tracked in the revised manuscript while the responses to your comments are as follows.

Lines 193-205: I think this is based on the bias of the sum or the mean over the two stations. But was the bias similar for both stations (in which case the bias correction is well justified), or was it much larger for one station than the other (in which case, the correction might be less justified by the data)?

Response: Thank you for the comment. As presented in following Figure 1, the difference of ratios between Clewiston and Belle weather stations is not significant. We thus think the correction is acceptable.

Figure 1. The ratios of SRNSRDB/SRstation from 2008 to 2016.

Line 249 – Do you mean ‘standard deviation’ rather than ‘standard error’? Using ‘standard deviation’ would make more sense to me as a measure of the variation of NDVI within the given window (whereas standard error would be a measure of the precision of the estimated mean NDVI within the window). Similarly for other uses of ‘standard error’ (e.g. line 474 and other places).

Response: Thank you for the suggestion. We have replaced “standard error” by “standard deviation”.

Eq 12: I found this equation a bit confusing – does it need brackets? (Probably should be around the ‘N-1’ , though I’m not sure why you use ‘N-1’ rather than ‘N’ as the denominator for RMSE).

Response: Thank you for the comment. We have added a bracket. ‘N-1’ was used to obtain the unbiased estimation of RMSE.

Fig 9: Planting area seems to be presenting in units of t/ha – I would have expected it to be in units of area, ie ha.

Response: This is the unit of yield. Therefore, ‘t/ha’ is presented in Figure 9.

Round 2

Reviewer 1 Report

Unfortunately, my opinion is different from the authors. The model can be evaluated without recalibrating the parameters and considering the recalibration, so the authors could verify the need for re-calibration for the applied region - note that this result applies only to the region where the authors are testing the model (if were applied to another region of the world the same procedure would be necessary). The model can be evaluated simply by separating the data into two sets, one for calibration and one for validation. In the way that the study was carried out, the conclusion of the paper is partial, that is, that some versions of the model can be directly applied in re-calibration and others cannot, but recalibration would be recommended.

Author Response

Responses to Reviewer 1

Unfortunately, my opinion is different from the authors. The model can be evaluated without recalibrating the parameters and considering the recalibration, so the authors could verify the need for re-calibration for the applied region - note that this result applies only to the region where the authors are testing the model (if were applied to another region of the world the same procedure would be necessary). The model can be evaluated simply by separating the data into two sets, one for calibration and one for validation. In the way that the study was carried out, the conclusion of the paper is partial, that is, that some versions of the model can be directly applied in re-calibration and others cannot, but recalibration would be recommended.

Response: Thank you very much for the comment. As suggested by the reviewer, recalibration (two thirds of data for calibration and one third of data for validation in Palm Beach County) was conducted in the revised manuscript (please see section 4.2.3). We agree with the reviewer that the recalibration results improve the reasonability of the conclusion.

Reviewer 2 Report

I have just noted a couple of small corrections to be made in the attached manuscript. 

Author Response

Responses to Reviewer 2

I have just noted a couple of small corrections to be made in the attached manuscript.

Response: Thank you for the comments and suggestions in the attached file. We have revised our manuscript according to them.
